# Kangaroo mother care prior to clinical stabilisation: Implementation barriers and facilitators reported by caregivers and healthcare providers in Uganda

**Victor S. Tumukunde**[1,2]*, **Joseph Katongole**[1], **Stella Namukwaya**[1], **Melissa M. Medvedev**[2,3,4], **Moffat Nyirenda**[1,2], **Cally J. Tann**[2,3,5], **Janet Seeley**[1,6], **Joy E. Lawn**[2,4]

1 Medical Research Council/Uganda Virus Research Institute and London School of Hygiene & Tropical Medicine Uganda Research Unit, Entebbe, Uganda, 2 Faculty of Epidemiology and Population Health, London School of Hygiene & Tropical Medicine, London, United Kingdom, 3 Maternal, Adolescent, Reproductive & Child Health Centre, London School of Hygiene & Tropical Medicine, London, United Kingdom, 4 Department of Paediatrics, University of California San Francisco, San Francisco, California, United States of America, 5 Department of Neonatal Medicine, University College London, London, United Kingdom, 6 Faculty of Public Health and Policy, London School of Hygiene & Tropical Medicine, London, United Kingdom

* Victor.tumukunde@mrcuganda.org

## Abstract

Kangaroo mother care (KMC) is an evidence-based method to improve newborn survival. However, scale-up even for stable newborns has been slow, with reported barriers to implementation. We examined facilitators and barriers to initiating KMC before stabilisation amongst neonates recruited to the OMWaNA study in Uganda. The OMWaNA study was a randomised controlled trial that examined the mortality effect of KMC prior to stabilisation amongst newborns weighing ≤2000 grams. At the four trial hospitals, we conducted focus group discussions (FGD) separately with caregivers and healthcare providers, in-depth interviews (IDI) with caregivers and key informant interviews (KII) with hospital administrators and healthcare providers. The World Health Organisation (WHO) Health Systems Building Blocks were used to guide thematic analysis. Eight FGDs (4 caregivers, 4 healthcare providers), 41 caregiver IDIs (26 mothers, 8 grandmothers, 7 fathers), and 23 KIIs were conducted. Key themes based on the building blocks were; family and community support/ involvement, health workforce, medical supplies and commodities, infrastructure and design, financing, and health facility leadership. We found that the presence of a family member in the hospital, adequate provision of healthcare workers knowledgeable in supporting KMC prior to stability, and adequate space for KMC beds where neonatal care is being delivered, can enable implementation of KMC before stability. Implementation barriers included fear of inadvertently causing harm to the newborn, inadequate space to practice KMC in the neonatal unit, and a limited number of trained healthcare workers coupled with insufficient medical supplies.

**Data Availability Statement:** The data used in this paper are included in S1 File.

**Funding:** Funded by the Global Health Trials scheme of the Department of Health and Social Care, the Foreign, Commonwealth and Development Office, the Medical Research Council, and the Wellcome Trust (MR/S004971/1 awarded to J.E.L.). A grant from the Eunice Kennedy Shriver National Institute of Child Health and Human Development of the National Institutes of Health (K23HD092611 awarded to M.M.) also supported this work. The content is solely the responsibility of the authors and does not necessarily represent the official views of the National Institutes of Health or other institutions with which the authors are affiliated. The funders had no role in study design, data collection and analysis, decision to publish, or preparation of the manuscript.

**Competing interests:** The authors have declared that no competing interests exist.

## Introduction

Globally, an estimated 2.3 million neonatal deaths occurred in 2022 [1], with over 80% occurring in small newborns, primarily due to being born preterm (<37 weeks' gestation) and/or small-for-gestational age [1, 2]. Major mortality reductions could be achieved by improving facility-based care of small and vulnerable neonates in low- and middle-income country (LMIC) settings [3–5]. Kangaroo mother care (KMC), as a component of small and sick newborn care, is associated with decreased mortality among clinically stable newborns [6–8]. However, there are several barriers to KMC implementation both within and outside the healthcare system [9].

KMC is a package including early, continuous, and prolonged skin-to-skin contact (SSC), usually with the mother, and promotion of exclusive breastmilk feeding, and is associated with early hospital discharge, requiring adequate support and close follow-up at home [10]. KMC has been found to be associated with decreased mortality, sepsis, hypothermia, hypoglycaemia, and length of hospital stay in clinically stable neonates [6]. KMC has multiple benefits for the neonate and their family, including improved duration of breastfeeding and growth outcomes [11, 12], supporting maternal-infant bonding and improved maternal mood [13].

A WHO-led trial recently reported a 25% reduction in mortality within 28 days among neonates born weighing 1000–1799 grams (g) who received KMC immediately after birth relative to those who received standard neonatal intensive care, including KMC after stabilisation [14]. In November 2022, the WHO released updated guidelines recommending that KMC should be initiated as soon as possible after birth, before stabilisation, in all preterm or low birthweight (LBW, <2500g) newborns, and given for 8–24 hours per day [15].

Despite clear evidence of impact in improving survival among newborns [6], scale-up of KMC, even for stable newborns, has been slow, hindered by a lack of national investment to implement at scale, and hard to track given lack of coverage data in routine health information systems [16, 17]. The majority of mothers of LBW newborns in many studies report being committed to KMC, satisfied with the results of weight gain and interested in continuing KMC at home [18]. However, barriers to KMC provision amongst stable newborns occur at both the health facility and community levels [9, 19, 20]. Studies have found that a lack of beds and space, privacy issues, inadequate caregiver education, insufficient staff and monitoring devices, and difficulties ensuring many hours of KMC, were common barriers to high-quality KMC practice in facilities [21–26]. Maternal factors, such as fatigue, depression, and postpartum pain, especially after a caesarean section, may reduce uptake and especially duration [9, 27, 28]. Providing continuous and extended KMC has been reported as a barrier for mothers and their families, due to competing household and work responsibilities, lack of money for transportation, and time required for commuting [28, 29]. At the community level after discharge, barriers to KMC adoption and continuity include stigma and guilt related to having a preterm newborn, gender roles in childcare, and adherence to traditional newborn practices [21, 29–31].

Most published studies on KMC barriers and facilitators have been carried out amongst stable newborns. Yet, in the wake of the updated WHO recommendations regarding KMC [15], there is need to understand the facilitators and barriers of KMC implementation prior to clinical stability. One study that examined the feasibility of KMC amongst newborns (≤2000g) before stability in a level-2 neonatal unit in Uganda found that the intervention was acceptable, but healthcare providers and parents expressed reservations concerning neonatal monitoring and space availability [21].

OMWaNA (Operationalising kangaroo Mother care amongst low birth Weight Neonates in Africa) was a randomised controlled trial, which recruited 2221 neonates ≤2000g in five government hospitals in Uganda, that examined the effect of KMC 'prior to stability' on mortality and morbidity outcomes [32]. This trial offered an important opportunity to explore, for

the first time, barriers and facilitators of KMC implemented prior to stability amongst small vulnerable newborns [32]. In this paper, we examine the barriers and facilitators to initiating KMC among neonates before stabilisation from different stakeholders' perspectives.

## Methods

### Study design and setting

This was a qualitative study embedded within the OMWaNA trial [32]. This randomised trial was conducted in the WHO level-2 neonatal units of five government hospitals in Uganda from 09-10-2019 to 31-07-2022. The participating hospitals were Entebbe Regional Referral Hospital, Jinja Regional Referral Hospital, Masaka Regional Referral Hospital, Iganga District Hospital, and Kawempe National Referral Hospital. Entebbe Hospital was not utilised for this sub-study because of early discontinuation due to becoming the national Covid-19 referral hospital and the neonatal unit being restricted. The trial recruited neonates weighing 700-2000g prior to clinical stability, which was defined as receiving ≥1 of the following treatments: oxygen support, continuous positive airway pressure (CPAP), intravenous (IV) fluids indicated for neonates unable to take enteral feeds, therapeutic antibiotics, and phenobarbital medication used to treat neonatal seizures. Pragmatic infrastructure improvements were undertaken prior to trial initiation, to allow for caregivers' beds in the neonatal care units, enabling safe implementation of the intervention of KMC prior to clinical stability [33].

### Participant recruitment

Caregivers were recruited from both the intervention arm and the control arm. Caregivers were identified among those whose neonate was either still admitted or already discharged with available phone contacts. Identified participants were contacted and given an appointment to return to the hospital for the interviews. The selected healthcare worker (HCW) participants included the research assistants for the OMWaNA trial and the government HCWs working in the neonatal care units who were on duty at the time of the interview or willing to come to the hospital on their off-duty days. Research assistants had been trained in KMC prior to stability and were familiar with the practice which led to more candid and informed responses as opposed to responses referring to conventional KMC. The hospital administrators were found in their offices on the appointed days for the interviews. Written informed consent was first sought from all participants. An impartial witness was used during the consenting of non-literate caregivers as per the ICH-GCP (International Council for Harmonisation Good Clinical Practice) guidelines. The ethical approval was obtained from Uganda Virus Research Institute ethics committee, the Uganda National Council for Science & Technology (UNCST) and the London School of Hygiene and Tropical Medicine Research Ethics Committee.

### Inclusivity in global research

Additional information regarding the ethical, cultural, and scientific considerations specific to inclusivity in global research is included in the S1 Checklist.

### Data collection and management

Focus group discussions (FGD) were held with caregivers, and separately with healthcare providers, at four of the OMWaNA trial hospitals. In addition, in-depth interviews (IDI) with caregivers and key informant interviews (KII) with neonatal healthcare providers and hospital administrators were conducted. Individual interviews allowed respondents to share personal

experiences, views and stories, while group discussions offered the opportunity for small groups of respondents to share ideas, serving to stimulate discussion amongst those present. The interviews were conducted by a trained social science research assistant and the lead investigator, and audio-recorded. In the event a participant declined the use of the audio-recorder, field notes were recorded, and the interview was transcribed from those notes. FGDs, IDIs and KIIs were conducted in Luganda, Lusoga, and English languages using structured topic guides, which were piloted on 4 caregivers and 3 nurses from one hospital. Data were collected until the team could not identify new themes emerging from the FGDs and interviews. In total, 4 healthcare worker FGDs, 4 caregiver FGDs, 41 caregiver IDIs, 19 healthcare worker KIIs (paediatricians, medical officers, nurses), and 4 hospital administrator KIIs were conducted.

All data were stored on secure servers at the MRC/UVRI & LSHTM Uganda Research Unit. The transcripts and audio files were stored on a computer as password-protected files. No names or other direct identifiers appeared in the transcripts or recordings; unique identification codes were used to ensure confidentiality.

## Data analysis

Data were analysed using a thematic content approach. The analysis involved familiarisation with the data, identifying codes and themes, developing a coding framework, and applying it to all of the data. The principal investigator and the social scientist read all the transcripts, generated themes, and developed the final coding framework. Transcripts were coded and managed using Nvivo 12.

## Theoretical framing

The WHO Building Blocks for Strengthening of Health Systems [34] have been adapted and applied to the provision of small and sick newborn care (SSNC). This adapted version describes the health system in terms of seven components (building blocks), notably; family centred care at the core then infrastructure, leadership/governance, financing, health workforce, medicines/supplies, and health information systems [35, 36]. We utilised these adapted building blocks for SSNC as a framework to inform our thematic analysis [35]. These building blocks align to WHO norms for care of small and sick newborns [15] and also to the more recent WHO/UNICEF ten core components of SSNC, of which seven are WHO Health Systems Building Blocks [37]. However, the health information systems building block was not linked to the themes generated from the interviews. Therefore, only six components were used. We mapped our generated themes to this version of the conceptual framework to guide our analysis of facilitators and barriers to KMC implemented prior to stability. Fig 1 is the conceptual framework for schematic analysis of barrier & facilitators of KMC by six adapted health system building blocks. In the results section, we present the barriers and facilitators by each main theme in turn.

## Results

The FGDs, IDIs, and KIIs, were conducted across the four hospitals, as indicated in Table 1. The barriers and facilitators were categorised under five themes, as described above: i) family and community support and involvement; ii) health workforce; iii) medicines and supplies; iv) infrastructure; v) health facility leadership. The main themes and sub-themes are summarised in Table 2.

### Family and community support and involvement

**Facilitators.** Family and community support and involvement referred to the perception and reality that the caregiver's family members assisted her or him to perform KMC. This

**Family & community support & involvement**
- Family member present in the hospital
- Cultural beliefs & practices

**Human resources for Health**
- Staffing levels & competency
- Ability to support caregivers

**Medical supplies & devices**
- Availability of medicines & devices
- Timely treatment

**Infrastructure and design**
- KMC space including in high dependency area, maternal beds, and privacy.
- Provision of washrooms for families

**Financing**
- Adequate financing to enable high coverage and quality of care.

**Health facility leadership**
- Support for staffing & training
- Overall enabling policy environment, with accountability

**Fig 1. Conceptual framework for schematic analysis of barrier & facilitators of KMC, by six adapted health system building blocks.**

support in the care of newborns had a positive impact on the continuity of KMC practice. Family members, especially fathers, provided financial support, and grandmothers and sisters helped with chores in the hospital such as washing clothes and going out to buy necessities. Family members who had had experience of taking care of a preterm infant previously encouraged caregivers to perform KMC. The positive attributes of family support were mentioned by both the HCWs and caregivers, especially mothers. One mother, for example, mentioned the food which family members brought as very helpful, and confidence in financial support once home with the new baby, kept her focused on performing KMC. Caregivers were grateful to have someone as a substitute KMC provider such as grandmothers, husbands, and sisters. One

**Table 1. Participants interviewed from different hospitals.**

|  | Persons involved | Hospital-1 | | Hospital-2 | | Hospital-3 | | Hospital-4 | |
|---|---|---|---|---|---|---|---|---|---|
| Hospital type | | National | | Regional | | Regional | | District | |
| Study arm | | KMC | Control | KMC | Control | KMC | Control | KMC | Control |
| **Number of HCWs FGD** | | 1 | | 1 | | 1 | | 1 | |
| **Number of Caregiver FGD** | | 1 | | 1 | | 1 | | 1 | |
| **Caregiver IDIs** | Mothers | 3 | 2 | 3 | 4 | 3 | 2 | 4 | 5 |
| | Grand Mothers | 2 | 0 | 1 | 1 | 1 | 1 | 2 | 0 |
| | Fathers | 2 | 0 | 2 | 0 | 0 | 1 | 2 | 0 |
| **KII with HCWs** | Neonatal nurses | 2 | | 4 | | 2 | | 4 | |
| | Medical officer | 1 | | 0 | | 1 | | 1 | |
| | Paediatricians | 1 | | 1 | | 1 | | 1 | |
| | Administrators | 1 | | 1 | | 1 | | 1 | |

**Table 2. Overview of the barriers and facilitators of KMC prior to clinical stability.**

|  | Facilitators | Barriers |
|---|---|---|
| **Main themes** | **Sub-themes** | **Sub-themes** |
| Family & Community support & involvement | • Financial support from family member<br>• Presence of family member in the hospital<br>• Provision of meals for the KMC mothers | • Caregiver being alone in the hospital.<br>• Maternal morbidity<br>• No other person to do KMC.<br>• Fear of hurting the small baby<br>• Fear of handling the baby• Cultural beliefs |
| Health workforce | • Supportive & encouraging HCWs to caregivers.<br>• Availability of adequate number of trained HCWs in preterm care<br>• Educating caregivers on KMC practice and its benefits | • Lack of knowledge on benefits of KMC and how to perform it.<br>• Rude and unsupportive HCW<br>• Fear of unstable baby deteriorating |
| Medical supplies & devices | • Availability of medicine for care of sick newborns and timely treatment<br>• Availability of medical equipment for monitoring sick newborns<br>• Availability of adequate oxygen supply points | • Buying medicines from outside the hospital<br>• Sick newborns missing treatment |
| Infrastructure and design | • Availability of adequate space with privacy<br>• Adequate number of KMC beds<br>• Clean environment and bathrooms/toilets | Lack of privacyUnclean bathrooms<br>• Overcrowding |
| Health facility leadership | • Adequate staffing for KMC care<br>• Training of HCW in preterm care<br>• Clear KMC guidelines / policies | • Lack of involvement by the leadership |
| Financing | • Adequate finances to support recruitment of healthcare worker and infrastructure improvement. | • Not improving budgetary allocation to meet the demand |

mother described how she would ask the baby's father to help her by putting the baby into KMC. She went on to say, '*some days he does put them, and you have to thank God when he wears the Kangaroo wrap*'. One father also described his involvement in the care of the newborn in a positive light: "*I wouldn't refuse to do what I have been told to do to enable my baby to add weight because I also want him to gain weight*". Thus, the presence of a family member in the hospital to support the mother was fundamental for continuity of KMC prior to stability.

Both mothers and fathers were motivated and hopeful about KMC practice. During one FGD, a mother said, "*For you to benefit from the Kangaroo method, you have to be patient and be ready to face all the challenges and accept all what the health workers tell you to use or stop using*". The fathers who were interviewed expressed their determination to perform KMC following the explanation of the benefits. One father commented:

> "*The fact that I want my baby to be alive gives me hope and strength to do whatever I am told to do as long as it will help him. I have to follow what I am told so that I get what I want from it. I wouldn't refuse to do what I have been told to do to enable my baby to add weight because I also want him to gain weight.*"

One HCW also expressed the importance of supporting the mothers during KMC, as mentioned by one HCW: "*if there is a way of providing for the basic needs of mothers who are going to stay longer because their babies have not stabilised, it would help improve*".

**Barriers.** Some families lacked the finances to support the hospital stay, which undermined the continuity of KMC prior to stability. Given that most families depended on fathers as the source of finances for upkeep during the hospital stay, they played a role in decision making as elaborated in the quote by one mother below.

> "*The father of the baby had stopped giving us money for food, remember I was in ward eight for a whole month so when they told us to buy syrups for the baby, we called him because we*

*didn't have money anymore, but he told us that he didn't have money anymore and that he was tired of the situation even the money for food we didn't know where to get it from*".

In such situations of financial constraint, families wanted the mother and baby discharged quickly, as mentioned by one healthcare worker: *"they want to leave early because of their financial status"*. Poorer families may have wanted to support the mother but were unable to do so, thus impeding KMC continuity.

Other families actively discouraged KMC by advising caregivers to use alternative methods of warming preterm babies, such as using hot water in jerricans and using a charcoal stove under the bed: "*the community usually think of other methods like using a hot charcoal stove or putting jerry cans filled with hot water around the baby*", said one father. A more fundamental barrier to KMC was the belief that preterm infants cannot survive and that KMC is a waste of time. This belief led to some fathers withdrawing all the financial support for the upkeep of the mother and child during the hospital stay and demanding discharge. One healthcare worker said: "a *father seeing the baby on oxygen believes the baby is dead and demands for discharge*". However, some mothers got financial support from other sources rather than the fathers. These included other family members and credit facilities as mentioned in the quotes below.

*"A family member might come and give you 50,000/-, at that time, the focus immediately goes back to the child. You send the 20,000/- to help at home and you remain with the 30,000/- to help at the hospital" (Mother).*

*"The mother got a loan from a certain SACCO [Savings and Credit Cooperative Organisation] she belongs to. She has not paid them yet" (Father).*

The combination of lack of financial support, and inadequate knowledge on survival of preterm infants and how KMC benefits unstable small and sick newborns impeded KMC continuity greatly.

There were other cultural barriers to KMC, such as prohibitions on a woman's in-laws seeing her undressed, or a grandmother helping to practice KMC in front of her son-in-law. One father commented: *"the grandmother to the baby cannot undress and do kangaroo in the presence of the baby's father"*, and he went on to say that "*a mother cannot do KMC in the presence of her father in-law*". Also, beliefs surrounding the gender roles were mentioned, with one father saying that: *"the community can't imagine us men putting our babies in the chest for a while"* meaning that this is supposed to be done by the mother. This deprived the mother of the help of a substitute KMC provider in case of twins or maternal morbidity.

Fatigue, tiredness, and maternal morbidity especially after caesarean section were mentioned as some of the maternal concerns that limited KMC prior to stabilisation. This was described by one HCW: "*there is no one to do Kangaroo especially for a mother who has been operated upon and as a caretaker she is still weak and receiving medication*". One mother also described the difficulties related with the skin-to-skin position, *"I experienced back ache when I put the child in my chest while in one position facing up, I also get chest pain because of putting the child in the chest for long."* These maternal related factors that impeded KMC might be alleviated with the presence of a family member to help in performing KMC during the hospital stay.

## Health workforce

**Facilitators.** Healthcare workers' availability and ability to health educate, counsel, and encourage caregivers to perform KMC, facilitated its implementation prior to stability. One

female caregiver said, "*the healthcare workers made me get used [to Kangaroo]*. Another caregiver commented: *"the nurse told me that her last born was also a preterm baby and encouraged me that my baby will grow normally if I continue putting the baby in my chest daily."*

Enabling the caregivers to understand the advantages of KMC had a positive impact on the continuity of KMC practice. A male caregiver described the information he had received from a HCW:

> *She* [the healthcare worker] *told us to put the baby in the chest on bare skin and tie the baby to get warmth, that because this baby is still very young it still needs the mother's warmth so when you put the baby in the chest it will be as if they are in the womb. My wife started putting the baby and she also told me that even if we go back home, we should continue putting the baby in the chest even I the husband I should help my wife to put the baby in the chest and we did, we put the baby in the chest like for thirty minutes."*

This provision of clear information was helpful to the caregivers, and several caregivers expressed their appreciation for the information they were given.

Within the KMC ward, peer support from other mothers who shared their KMC experiences also promoted the practice. One mother described the encouragement she and the new mother had received when they were facing challenges associated with KMC.

> *"During that time people would come in and they had brought their babies for review, and they would tell us their stories and encouraged us to be strong that also their babies were like that, which made us believe."*

With such encouragement fears were alleviated, and caregivers began to practice KMC. One mother said: *"at first, I saw that it was a burden but after I believed that the preterm baby is my own and I want the baby to survive so I must do whatever they tell me"*.

Continuous counselling on the benefits of KMC and encouragement of caregivers to perform KMC by HCWs and peers was seen as a motivating factor.

**Barriers.** Low staffing levels of HCWs limited their capacity to monitor and support implementation of KMC prior to stabilisation, as described by one of the HCWs: *"shortage of staff makes monitoring hard because sometimes we are overwhelmed by the numbers"*. This in turn limits the healthcare worker's zeal to support the implementation of KMC prior to stabilisation.

HCWs mentioned that the lack of understanding of KMC procedures and its benefits by the caregivers was an impediment to KMC implementation. One HCW observed that: *"there is not enough education about what Kangaroo is"*. This led to poor practice of KMC with no skin-to-skin contact as described by another HCW: *"you find the mother dressed, and the baby is dressed while in KMC position"*. With low staffing levels and associated worker load, the HCWs are unable to support caregivers fully with health education and the necessary counselling, thus impeding the KMC practice in the hospital.

Fear by the healthcare workers that caregivers do not know how to do KMC and that they might harm the unstable baby was another hindrance. One HCW said that *"in my opinion, the other fear is lack of knowledge and skills gap [by caregivers] of doing KMC"*. She went on to give an experience of a fatal outcome: *"we got a preterm in that they pretended to put the baby in KMC but instead killed the baby [healthcare worker narrating how a caregiver performed KMC wrongly and the newborn died]"*. This fear was augmented by the fact that KMC prior to stability was a new intervention to the HCWs and they had no prior training in the procedure before the trial. This was particularly the case for the regular hospital staff who were not trial specific HCWs.

There were some HCWs who continued to hold negative attitude towards KMC due to a lack of adequate knowledge of KMC among unstable preterm babies. The HCWs need to be "*educated because not all of them know how to take care of the neonates, so they should be trained*" as expressed by one neonatal care unit in-charge. The caregivers interpreted the poor attitude of HCWs as poor 'customer' care which discouraged them from practicing KMC at the health facility. This was described by one mother that "*nurses should change, they reach a point where they shout at us yet even us, we are not yet used to the babies and still scared*".

In some instances, caregivers received conflicting instructions from the HCWs which is likely to be due to lack of training. For example, one mother said, "*a doctor tells you to give the baby 20 millilitres of breast milk but don't breast feed and another HCW comes and complain as to why you are not breast feeding the baby and tells you to breast feed*".

Healthcare workers' lack of knowledge on KMC prior to stability could be the cause of this communication gap between the HCWs and the caregivers, which discouraged the implementation of KMC prior to stabilisation.

## Medical supplies and commodities

**Facilitators.** The availability of medicine and timely treatment enabled the caregivers to perform KMC with the hope that the newborns would survive. This was described by one caregiver: "*the babies would receive treatment through cannulas all the time until we left when the babies had improved, they gave them good treatment*". This gave the caregivers enough time to perform KMC as they did not have to stop KMC to go and buy the medicines.

HCWs felt that having functioning equipment was an important facilitator for the care of small and sick newborns, "*we also need functional monitors because we need to monitor these babies*". The equipment enabled them to monitor the sick newborn's progress while in KMC.

**Barriers.** An inadequate supply of medicines and other essentials for small and sick newborn care was a barrier as described by one mother that "*Musawo* (healthcare worker) *might send you for medicine and you fail to get it*". This created uncertainty about the survival of the newborns and interrupted the skin-to-skin practice which demoralised the caregivers.

The lack of an adequate number of oxygen supply outlets on the ward hindered KMC prior to stability in one hospital, as babies had to be nursed from shared oxygen points away from KMC beds. This meant that newborns who needed oxygen support had to halt KMC and be transferred to oxygen outlet points as elaborated by one HCW: "*Sometimes when you initiate a baby on KMC, you find that we don't have oxygen and we have to use the few concentrators we have for all the babies*".

A lack of functioning equipment for monitoring the vital signs of sick newborns was also reported as a barrier. Without vital monitoring, the HCWs feared that the newborn might die unnoticed while in skin-to-skin position, as described by one HCW: "*you might find this baby in KMC, dead*".

There was also the reported fear by the HCWs and caregivers of disconnecting oxygen tubes and vital sign monitors from the newborns while performing KMC.

One mother expressed her fear of the machines:

"*Since it is the one helping my baby to breathe, what if you touch it yet you do not know how to put it right and instead you be the one to kill your own baby*". Another caregiver thought that "*the oxygen tubes would come out of the nose and the baby fails to breathe*". A grandmother concurred with this view: "*when the baby is on oxygen it is not possible to do KMC because the baby has tubes in the nose, how can you carry the baby without those tubes coming out?*". Some healthcare workers also feared that "*the mother might take-off the baby from*

*oxygen while putting the baby in the chest because the nurse has told them to put babies in kangaroo and baby might stop breathing*".

The fear of harming the baby is a crucial barrier to implementation of KMC prior to stability.

### Infrastructure and design

**Facilitators.**   The availability of space and beds for practicing KMC on wards was reported in one hospital to have encouraged the caregivers to continue with KMC prior to stability. Caregivers in that site reported privacy on the KMC ward as a motivator to continue with the KMC practice as compared to the general wards: "*every parent who is doing Kangaroo has their own bed even Nalongo [mother of twins] was given two beds and she has enough space to do Kangaroo.*" Room for KMC with beds and privacy for the mothers greatly facilitated implementation of KMC prior to stability.

**Barriers.**   The lack of cleanliness in the hospitals and the washrooms was another hindrance reported at two of the hospitals. The poor state of the toilets and the inability of the hospital to clean the gowns worn by the KMC mothers was reported as a barrier too. This was expressed by one caregiver: "*uncleanliness in your environment for example when you enter a dirty place, or you don't wash hands, but you come and touch the baby*" are some of the limiting factors for the continued stay in the hospital.

Inadequate space that led to overcrowding in the KMC ward and babies sharing incubators in instances where they had to return the babies to the warmer or incubator made the caregivers worry about the health of their newborns. Caregivers worried about the spread of infection through the sharing of medical devices and equipment. One caregiver described how her baby was sharing a vital sign monitoring machine with other babies, "*sometimes, they would remove it from my baby and put it on another baby, then bring it back after some time, they should increase on the amount of equipment at the hospital*".

One HCW intimated that "*we don't have space where to put these mothers to practice KMC*". Another HCW said that "*they would even tell us they want to go home since they have been discharged from the maternity ward and wondered why we can't discharge them*". The lack of adequate infrastructure hindered the continuity of KMC prior to stability.

**Health facility leadership.**   HCWs from one of the hospitals reported a lack of adequate involvement by hospital leadership which hindered full scale capacity building in terms of staff training and supply of other resources for KMC implementation. This was expressed by a HCW who rated hospital administration involvement to be "*almost 20% support and that is the provision of those four beds and the oxygen,*" and was quick to add that "*we actually need people to push them to understand KMC so that they can help us*".

**Financing.**   The hospital leadership we interacted with acknowledged the ever-increasing demand for neonatal care services amidst limited resources. The administrators in all four hospitals mentioned low staffing levels, limited space for adequate KMC ward, and an insufficient budget allocation; as described by one administrator, "*we are operating in a very old staff structure even when the numbers [of patients] have increased, if we have 30 midwives and three duties how many can work per shift?*" For implementation of KMC prior to stability to be successful, there is need for increased funding to support infrastructure improvement and increase the staffing levels especially for the nurses.

We have described above the identified facilitators and barriers of KMC prior to stabilisation, which are both for general KMC implementation and for specific KMC prior to stability. It is clear from our findings that the successful implementation of KMC depends on the

willingness of caregivers and HCWs to practice it, and the provision of adequate facilities in hospitals to support KMC.

## Discussion

This qualitative study presents insights for the implementation of KMC prior to stability from the perspective of the caregivers, healthcare providers, and hospital administrators in four neonatal care units in Uganda. This qualitative study was embedded in the OMWaNA trial, which recruited 2221 neonates and found significant changes in intermediary outcomes and a nonsignificant 14% effect on neonatal mortality [38]. WHO policy recommends implementation of KMC prior to stability and our paper gives new insights on implementation realities [15]. The important facilitators identified included: presence of a family member in the hospital as a substitute KMC provider, availability of vital signs monitoring devices, adequate oxygen supply points and adequate number of staff trained in KMC prior to stability. The important barriers included maternal morbidity, fear of hurting the unstable newborn, fear of disconnecting oxygen and unstable baby deteriorating, absence of supplies for care of sick newborn, healthcare workers' lack of knowledge in KMC prior to stability, lack of adequate space for KMC beds within the neonatal care unit, and inadequate involvement of the hospital leadership.

Family support and involvement as a facilitator of KMC has been reported in other studies among stable neonates which postulated that, competing activities like household chores are a hindrance to continuity of KMC [39]. In this study, we found that activities like taking care of the laundry during the hospital stay and going to buy food interfered with care. However, an important hindrance was the absence of medicines for SSNC necessitating caregivers to move outside of the health facility to buy these medicines specifically hindered the continuity of KMC prior to stability. The presence of a family member in the hospital to help with these errands allowed the mother to perform KMC continuously, a finding corroborated in the work of Smith et al [30]. In our study, other family members also helped in performing KMC in case of twins or maternal morbidity. Thus, the barrier to KMC prior to stability due to maternal morbidity, fatigue, and tiredness could be alleviated by allowing a family member to perform KMC on behalf of the mother. This facilitator has been reported in other studies at the community level and facility level where the sister, husband, or mother-in-law helped to perform KMC at home or at the health facility [21, 26]. Research findings also show that, the presence of a companion in the hospital facilitates promotion of KMC and strengthens moral support for the mothers [24]. In practical terms, facilitating KMC also requires space and washrooms and possibly financial support for families.

We found that cultural beliefs and practices were a barrier to immediate KMC practice echoing findings from studies of KMC among stable neonates [30, 40, 41]. Although health education for mothers on the benefits of KMC tended to improve on the acceptability of the practice [21, 30], mothers tend not to have enough power in their family to repudiate their family members' beliefs [42]. Given that parents of preterm babies desire more information than what is provided normally [43], investing in health education of these parents is likely to dispel the belief surrounding the survival of these preterm infants and thus promote KMC prior to stability. This is in line with the WHO recommendations that families of LBW infants should be given extra support to care for their infants. This support may include education, counselling, discharge preparation and peer support [15]. Such interventions should also target the older generations of the family unit that is, the mothers-in-law and grandmothers, who do not find KMC to be appropriate [44].

The low staffing levels and an insufficient number of trained HCWs in small and sick newborn care inhibits the implementation of KMC at the health facilities. This is because the

HCWs felt that they do not have enough time to educate caregivers on KMC and they believed that these babies required close monitoring which was hard given the low staffing levels. This has been described in another study in which nurses believed that supporting the mothers was important but a lack of staff could not allow them to provide such supportive care [28, 45, 46]. If the HCWs would choose peer mothers on the KMC ward to support the new mothers in doing KMC, this would increase on the information sharing and subsequent uptake of the intervention without straining the already over-worked HCWs. This is because, support from peer caregivers has demonstrated to be important in facilitating KMC practice [20, 47].

We found that mothers may fear the clinical environment especially the monitoring machines, oxygen, and IV lines. First-time mothers particularly fear to touch their own vulnerable babies, demonstrating the great need for support from the healthcare workers as reported in other studies [20, 45]. For the implementation of KMC prior to stability to succeed, there needs to be a trained HCW to champion the training of other HCWs and who will in turn support the caregiver trainings. This mirrors what was found by Lee et al [48], who observed that when a healthcare worker who served as a promoter of KMC in a facility leaves, it increases the difficulty of educating other staff members on the practice of skin-to-skin care [48]. With the training, the perceived fear of harming the baby by both healthcare workers and caregivers will likely be alleviated and hence facilitate KMC prior to stability as some studies have found that inadequate training in KMC led to nurses having conflicting knowledge of duration of skin-to-skin contact and are hesitant to use KMC for infants with catheters, whether intravenous or umbilical [49]. There were similar findings in this study where caregivers received conflicting instructions from the HCWs, which was likely due to knowledge gap in care of small and sick newborns.

Space for KMC beds with privacy and clean sanitary facilities was found in our study to be a motivator for continuity of KMC prior to stability. Crowding and insufficient space has been a cause of hastened discharge from the health facilities with conventional KMC [20, 50], coupled with a lack of privacy and discomfort with being undressed in the presence of strangers [27, 51]. For the implementation of KMC prior to stabilization to occur, infrastructure remodelling is crucial to create space that can accommodate adult beds within the neonatal care units with good sanitary facilities for caregivers [33]. In addition, these facilities require an adequate number of oxygen supply points to support unstable newborns during KMC.

Support from the hospital management in terms of adequate staff allocation, training, availability of medical supplies and treatment guidelines was described as an important facilitator of KMC prior to stability. The same was reported by Yue et al [40] that provision of necessary equipment, KMC rooms and identification of nurses to specialize in KMC coupled with routine supervision from hospital management, was an incentive for the HCWs to continue implementing KMC [20, 40, 52]. The same is likely to facilitate the implementation of KMC prior to stabilization. However, the hospital managers blame the lack of resources that have failed them to recruit more HCWs and improve infrastructure as the main hindrances to KMC implementation.

## Strength and limitations

The main strength of the study is that it has drawn on the rich body of primary qualitative data using KIIs, IDIs, and FGDs, providing information that can help policymakers in the implementation of the new WHO guidelines that recommend immediate KMC. This is the first multi-site study to publish on barriers and facilitators of KMC prior to stability from the perspective of the HCWs and the caregivers. The study covered four health facilities at different levels of care (national referral, regional referral, and district hospital), which increases the

generalizability of the findings. The limitations of this study are that we had a limited number of male caregivers (fathers) participating and thus we could not get their views on paternal involvement in care of preterm babies, we sampled on the basis of availability which introduced some bias in our sampling. In addition, we were not able to capture the detailed demographic description of the respondents for further categorization of the responses. The involvement of research assistants as participants in the key informant interviews could have unintentionally generated biased responses.

## Conclusion and recommendations

The presence of a family member as a substitute KMC provider, recruitment, and training of healthcare workers in care of small and sick newborns, who will in turn support the families through education and counselling on KMC prior to stability are crucial for implementation of immediate KMC. In Tanzania, a national investment case for neonatal care has been transformative for additional funding for neonatal units including space, staff, and devices [53]. Additional resource mobilisation will be foundational, for implementing KMC prior to stability with respect for families and health workers, notably for medicines and supplies, and infrastructure changes to accommodate mothers' beds for continuous KMC.

## Supporting information

**S1 Checklist. Inclusivity in global research.**
(DOCX)

**S1 File. Data summary.**
(DOCX)

## Acknowledgments

Most importantly, we thank the mothers, newborns, and families who participated in this study. We give huge appreciation to the neonatal unit nurses, doctors, and staff at Iganga, Jinja, Kawempe, and Masaka Hospitals for their participation and dedication throughout the study. We also thank Salama Kasoga for assisting with data collection.

## Author Contributions

**Conceptualization:** Victor S. Tumukunde.

**Data curation:** Victor S. Tumukunde, Joseph Katongole, Stella Namukwaya.

**Formal analysis:** Victor S. Tumukunde, Joseph Katongole, Stella Namukwaya, Cally J. Tann, Janet Seeley.

**Funding acquisition:** Melissa M. Medvedev, Joy E. Lawn.

**Methodology:** Janet Seeley.

**Supervision:** Melissa M. Medvedev, Moffat Nyirenda, Cally J. Tann, Janet Seeley, Joy E. Lawn.

**Writing – original draft:** Victor S. Tumukunde.

**Writing – review & editing:** Melissa M. Medvedev, Moffat Nyirenda, Cally J. Tann, Janet Seeley, Joy E. Lawn.

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
