## [Decision Letter · Decision Letter 0]

28 Feb 2024

PGPH-D-23-02561

Kangaroo Mother Care prior to clinical stabilisation: Implementation barriers and facilitators reported by caregivers and health care providers in Uganda.

Dear Dr. Tumukunde,

Thank you for submitting your manuscript to PLOS Global Public Health. After careful consideration, we feel that it has merit but does not fully meet PLOS Global Public Health’s publication criteria as it currently stands. Therefore, we invite you to submit a revised version of the manuscript that addresses the points raised during the review process.

We look forward to receiving your revised manuscript.

Kind regards,

Anteneh Asefa Mekonnen, Ph.D.

Academic Editor

Journal Requirements:

1. Please ensure you have included the registration number for the clinical trial referenced in the manuscript.

2. Please include a complete copy of PLOS’ questionnaire on inclusivity in global research in your revised manuscript. Our policy for research in this area aims to improve transparency in the reporting of research performed outside of researchers’ own country or community. The policy applies to researchers who have travelled to a different country to conduct research, research with Indigenous populations or their lands, and research on cultural artefacts. The questionnaire can also be requested at the journal’s discretion for any other submissions, even if these conditions are not met.  Please find more information on the policy and a link to download a blank copy of the questionnaire here: https://journals.plos.org/globalpublichealth/s/best-practices-in-research-reporting. Please upload a completed version of your questionnaire as Supporting Information when you resubmit your manuscript.

Additional Editor Comments (if provided):

Reviewers' comments:

Reviewer's Responses to Questions

**Comments to the Author**

1. Does this manuscript meet PLOS Global Public Health’s publication criteria? Is the manuscript technically sound, and do the data support the conclusions? The manuscript must describe methodologically and ethically rigorous research with conclusions that are appropriately drawn based on the data presented.

Reviewer #1: Yes

Reviewer #2: Yes

2. Has the statistical analysis been performed appropriately and rigorously?

Reviewer #1: N/A

Reviewer #2: N/A

3. Have the authors made all data underlying the findings in their manuscript fully available (please refer to the Data Availability Statement at the start of the manuscript PDF file)?

Reviewer #1: Yes

Reviewer #2: Yes

4. Is the manuscript presented in an intelligible fashion and written in standard English?

Reviewer #1: Yes

Reviewer #2: Yes

5. Review Comments to the Author

Reviewer #1: Strong research design and well written, and it is coherent. The research used a qualitative inquiry and presented insight from the perspective of families/caregivers, health care workers and facility mangers and report important insight to barriers and facilitators to deliver kangaroo mother care in medically unstable state. I suggest the following revisions:

- In methods authors could describe how the adequacy of sample size determined. It is not clear how the adequacy of reported sample size is determined.

- In the result and discussion section more insight on highlighting financing issues-results from both families and health facility mangers reported as important barriers or enablers. For example, inadequate number of staffing, space, equipment could be due to low budget allocation. The opportunity cost to families coupled with fact families must buy medicine, food and other supplies is leading to increasing out of pocket expenses leading to catastrophic family expenses (as reported by other authors). The authors need to further examine the data.

- If the authors agree with my suggestion above, I suggest revising the conclusion and recommendation section to highlight adequate financing beyond infrastructure remodeling.

Reviewer #2: Good study.

The abstract as it will be available separately should have no abbreviations such as IDI, KIIs

You included the research assistants in the focus group which is concerning. They were part on the implementing team.

Not sure how the reader is expected to interpret focus group discussions separately from the IDI. For a non social scientists reading the study, this should be clarified in the methods.

Table 1- first time we see a control group and not sure how the control group comments were dealt with. Were they analyzed separately, were they included because you needed the numbers or some other explanation.

In the results for example line 191 to 202 even though you are trying to tell reader what the Healthcare worker said, the quotes are from parents/mothers.

It is difficult to tease out the healthcare worker concerns and family concerns

Medical supplies again mothers quotes.

Table 2 is good but you do not flow the quotes according to your table of themes.

In the discussion you indicate cultural beliefs but I may have missed, I did not see these in the IDI.

The story is compelling and you have a lot of good quotes, however it becomes quite long and the message can be given in a shorter version that people will more likely read.

6. PLOS authors have the option to publish the peer review history of their article (what does this mean?). If published, this will include your full peer review and any attached files.

**Do you want your identity to be public for this peer review?** For information about this choice, including consent withdrawal, please see our Privacy Policy.

Reviewer #1: **Yes: **Tedbabe Degefie Hailegebriel

Reviewer #2: No

---

## [Decision Letter · Decision Letter 1]

23 May 2024

PGPH-D-23-02561R1

Kangaroo Mother Care prior to clinical stabilisation: Implementation barriers and facilitators reported by caregivers and health care providers in Uganda.

Dear Dr. Tumukunde,

Thank you for submitting your manuscript to PLOS Global Public Health. After careful consideration, we feel that it has merit but does not fully meet PLOS Global Public Health’s publication criteria as it currently stands. Therefore, we invite you to submit a revised version of the manuscript that addresses the points raised during the review process.

Please make it clear in the Methods that the presence of researchers as participants enhances the credibility and trustworthiness of your study, given its qualitative nature. However, please also include a sentence or two about the limitations of potential biases that may have been introduced by having the research assistants as participants. 

We look forward to receiving your revised manuscript.

Kind regards,

Anteneh Asefa Mekonnen, Ph.D.

Academic Editor

Journal Requirements:

Additional Editor Comments (if provided):

Thank you for the revisions. Please make it clear in the methods that the presence of researchers as participants enhances the credibility and trustworthiness of your study, given its qualitative nature. However, please also include a sentence or two about the limitations of potential biases that may have been introduced by having the research assistants as participants.

Reviewers' comments:

Reviewer's Responses to Questions

**Comments to the Author**

1. If the authors have adequately addressed your comments raised in a previous round of review and you feel that this manuscript is now acceptable for publication, you may indicate that here to bypass the “Comments to the Author” section, enter your conflict of interest statement in the “Confidential to Editor” section, and submit your "Accept" recommendation.

Reviewer #1: All comments have been addressed

2. Does this manuscript meet PLOS Global Public Health’s publication criteria? Is the manuscript technically sound, and do the data support the conclusions? The manuscript must describe methodologically and ethically rigorous research with conclusions that are appropriately drawn based on the data presented.

Reviewer #1: Yes

3. Has the statistical analysis been performed appropriately and rigorously?

Reviewer #1: Yes

4. Have the authors made all data underlying the findings in their manuscript fully available (please refer to the Data Availability Statement at the start of the manuscript PDF file)?

Reviewer #1: Yes

5. Is the manuscript presented in an intelligible fashion and written in standard English?

Reviewer #1: Yes

6. Review Comments to the Author

Reviewer #1: the authors have addressed my minor comments. The manuscript is clear and logical.

7. PLOS authors have the option to publish the peer review history of their article (what does this mean?). If published, this will include your full peer review and any attached files.

**Do you want your identity to be public for this peer review?** For information about this choice, including consent withdrawal, please see our Privacy Policy.

Reviewer #1: **Yes: **Tedbabe Degefie Hailegebriel

---

## [Editor Report · Decision Letter 2]

13 Jun 2024

Kangaroo Mother Care prior to clinical stabilisation: Implementation barriers and facilitators reported by caregivers and health care providers in Uganda.

PGPH-D-23-02561R2

Dear Dr Tumukunde,

We are pleased to inform you that your manuscript 'Kangaroo Mother Care prior to clinical stabilisation: Implementation barriers and facilitators reported by caregivers and health care providers in Uganda.' has been provisionally accepted for publication in PLOS Global Public Health.

Best regards,

Anteneh Asefa Mekonnen, Ph.D.

Academic Editor